# Sequential Drug Delivery in Targeted Cancer Therapy

**DOI:** 10.3390/pharmaceutics14030573

**Published:** 2022-03-05

**Authors:** Han Yu, Na Ning, Xi Meng, Chuda Chittasupho, Lingling Jiang, Yunqi Zhao

**Affiliations:** 1College of Science and Technology, Wenzhou-Kean University, Wenzhou 325060, China; yhan@kean.edu (H.Y.); ningn@kean.edu (N.N.); 2Wenzhou Municipal Key Laboratory for Applied Biomedical and Biopharmaceutical Informatics, Wenzhou-Kean University, Wenzhou 325060, China; shjust@163.com; 3Faculty of Pharmacy, Chiang Mai University, Chiang Mai 50200, Thailand; chuda.c@cmu.ac.th

**Keywords:** sequential targeted, drug delivery, nucleus targeting, mitochondria targeting, cancer treatment

## Abstract

Cancer is a major public health problem and one of the leading causes of death. However, traditional cancer therapy may damage normal cells and cause side effects. Many targeted drug delivery platforms have been developed to overcome the limitations of the free form of therapeutics and biological barriers. The commonly used cancer cell surface targets are CD44, matrix metalloproteinase-2, folate receptors, etc. Once the drug enters the cell, active delivery of the drug molecule to its final destination is still preferred. The subcellular targeting strategies include using glucocorticoid receptors for nuclear targeting, negative mitochondrial membrane potential and N-acetylgalactosaminyltransferase for Golgi apparatus targeting, etc. Therefore, the most effective way to deliver therapeutic agents is through a sequential drug delivery system that simultaneously achieves cellular- and subcellular-level targeting. The dual-targeting delivery holds great promise for improving therapeutic effects and overcoming drug resistance. This review classifies sequential drug delivery systems based on final targeted organelles. We summarize different targeting strategies and mechanisms and gave examples of each case.

## 1. Introduction

Cancer is a devastating health issue leading to high mortality rates worldwide with a complex pathophysiology. Conventional therapies, including chemotherapy, radiation therapy and surgery, have been explored to eliminate cancer cells and increase the survival rates of patients [1]. However, these traditional treatments are not sufficient to eradicate all cancer cells with many limitations such as cytotoxicity, multi-drug resistance and poor selectivity [2]. In addition, the non-specificity often causes significant damage to healthy cells, and the lack of targeting potential resulting in low tumor accumulation leads to a relatively low therapeutic efficiency [2,3]. Therefore, it is desirable to develop targeted drug delivery systems that can deliver the therapeutic agent to the target tumor to reduce the adverse effects and improve efficacy.

Today, nanotechnology-based drug delivery systems for targeted cancer therapy have attracted increasing attention from researchers due to their ability to deliver therapeutic agents precisely to tumor sites [4,5,6,7]. The targeting strategies can be classified into passive and active targeting [8]. Passive targeting relies on the accumulation of the nanoparticles with specific physicochemical properties (size, charge, etc.) in the tumor tissue. This accumulation is due to the enhanced permeability and retention (EPR) effect and the particular tumor microenvironment [9]. However, the EPR effect was discovered in mice tumor-bearing models. Clinical studies indicated that the EPR effect does not work in humans [10,11]. Therefore, a novel targeting strategy in cancer treatment is in urgent need. Compared with the passive approach, active targeting can significantly increase the efficiency of the drug delivered to the desired tissues/cells via specific interactions between a targeting moiety on the nanoparticles and cellular/subcellular markers [12]. At the cellular targeting level, by conjugation of targeting moieties on the surface of nanoparticles, they could specifically target cancer cells with overexpressed receptors and increase cellular uptake and improve drug delivery efficiency [13]. For the subcellular level, researchers mainly focused on the organelle-targeted therapy that can further activate specific cellular pathways, enabling them to kill cancer cells [14,15,16].

Although the active targeted delivery systems in cancer therapy have been extensively reviewed elsewhere [17,18,19,20], few of them focused on sequential drug delivery that can target both cellular and subcellular simultaneously. Therefore, this review highlights the application of cell organelle sequential targeted drug delivery systems in cancer treatments depending on the different types of cellular and subcellular targeting moieties. Some specific characteristics of sequential-targeting drug delivery systems in cancer therapies are summarized in Table 1.

## 2. Nuclear Targeting

DNA is the genetic material of eukaryotic cells. It can directly influence the transcription and translation process in the cell. The translation and transcription products coordinate and control the physiological activities in cells and organisms. Many anticancer drugs are designed to destroy the cancer cells’ genes to inhibit their proliferation, which means that these drugs have to play their roles after entering the nucleus of cancer cells [27]. In this case, cancer cell destruction can be done by destroying its DNAs. The applications of nuclear-targeting drug delivery systems (DDSs) have received much attention because of higher curative efficiency, especially for cancer treatment [27]. Using methods to specifically target cancer cells and then target cancer cells’ nuclei and destroy the DNAs in them, we can kill the cancer cells and keep innocent cells alive. However, because of the compartmentalization in eukaryotic cells, therapeutic agent nuclear delivery efficiency is lower than the ideal model [51]. There are two major barriers that the drug has to overcome before it reaches the nucleus, i.e., cell membrane and nuclear membrane [52]. Endocytosis is the most common way for non-viral vectors to enter the cells [52]. There are two forms of endocytosis. First, the cell swallowed the vector as the foreign matter [24]. Second, the targeting moiety on the carrier surface can specifically bind to the receptor on the targeted cell membrane and then be swallowed by the cell through endocytosis [23]. After the nanoparticles enter the cytoplasm, some weakly basic molecules, such as polyamines, are able to burst the endosome by the “Proton Sponge Effect” [53].

In addition, the unique properties of the drug carrier surface can help the vectors straightly go through the cell membrane. For example, the hydrophobicity of TiO_2_-immobilized dopa-decyl that can penetrate cell membrane [54] and nucleolin’s active transport property of aptamer on liposome surface [55]. From the cell membrane to the nucleus, active nuclear targeting is needed to avoid being swallowed by lysosomes that may digest the drug. Some drug carriers contain zwitterionic or other pH-sensitive components on the surface. They are intended to control drug release in the nucleus by taking advantage of different pH values in the cytoplasm and near the nucleus [22,27,56,57]. Some anticancer drugs, such as doxorubicin (DOX) [21] and paclitaxel (PTX) [58], which are released in the cytosol, can target the nucleus by themselves. Other drugs can be driven by nucleus targeting moieties and then transported to the nucleus (Figure 1).

### 2.1. Peptide/Protein-Based Nuclear Targeting

Peptides and proteins can be used as a drug carrier or targeting moiety in nuclear targeting drug delivery. As the drug delivery carrier, protein can function like armor that protects the encapsulated drug while traveling to the targeted cell, and some proteins may have lower cytotoxicity [51]. Some peptides are used to modify the drug carrier surface. The functions of those peptides are cellular or subcellular targeting, which can selectively deliver the drug molecule to the targeted cellular component [21]. It is worth mentioning that peptide is sensitive to pH. Therefore, protein or peptides can transform their conformation due to differential pH in cancer cells [22].

For instance, the engineered modular DNA carrier (MDC) proteins were used as protein-DNA nanoparticle vectors, which contained the sperm chromatin component protamine for cellular-level targeting, diphtheria toxin’s endosome-translocation domain, and a sequence for optimizing nucleus localization. Inside of the cancer cell, MDC interacted with cellular nuclear transport proteins and actively trafficked to the nucleus to play its role [51]. However, it is more common to see proteins on the carrier surface as a targeting moiety. A novel cerasome nanoparticle targeted to programmed death-ligand 1 (PD-L1) was decorated with PD-L1 monoclonal antibody, targeting PD-L1 positive cancer cells. This strategy could elevate therapeutic responses to cancer immunotherapies. The cerasome nanoparticle loading paclitaxel was labeled with IRDye800CW and the MRI contrast agent, Gd-DOTA, to form PD-L1-PCI-Gd. The result showed that PD-L1-PCI-Gd could be used as a noninvasive in situ tumor imaging system. Due to its good fluorescent properties, the magnetic/fluorescent dual-mode nanoparticle could provide high spatial information, high T1 relaxivity, and high sensitivity [58]. In another study, a DOX-loaded liposome surface was modified with AR-CPP octaarginine (R8) and transferrin (Tf) (dual DOX-L) to target ovarian carcinoma A2780 cells. The targeted cells over-expressed transferrin receptors on the cell membranes, so the liposomes showed to enter the cell by receptor-mediated endocytosis. Then, DOX was actively delivered to the nucleus [21]. Additionally, Han et al. formulated pH-responsive core−shell structured nanoparticles (CSNPs) that consist of cationic core and anionic shell. The cationic core was made up of amino-functionalized mesoporous silica nanoparticles (MSN), modified with acid-cleavable polyethylene glycol (PEG) and TAT peptide. The anionic shell was constituted by galactose-modified poly (allylamine hydrochloride)-citraconic anhydride, a hepato-carcinoma-targeting polymer with the charge-reversible property. If the intercellular pH becomes more acidic, the anionic shell will converse to a positive charge and disassemble with the core. In this case, the TAT peptide could deliver DOX to the target nucleus [22].

### 2.2. Small Molecule Nuclear Targeting

Some small molecules such as dexamethasone and 10-hydroxycamptothecine (HCPT) can target specific cells or the subcellular parts inside the cell [23,24]. They can be modified on the surface of the nuclear targeting drug delivery system and help them localize or enter the target cells or the subcellular structure. A versatile shell-cross-linked nanoparticle (SCNP), which consisted of triblock zwitterionic copolymers, polycarboxybetaine methacrylate-block-poly (N-(2-(2-pyridyl disulde) ethyl methacrylamide–block-poly(2-(diisopropylamino) ethyl methacrylate) (PCB-b-PDS-b-PDPA, also as PCSSD), combined with RGD (Arg-Gly-Asp) peptide and ultra pH-sensitive PDPA core, was used to deliver DOX to the nucleus. Since it had an ultra pH-sensitive PDPA core and the cytosol and near nucleus region have different pH values, the nanoparticle could change its assembly state when it moved to the nucleus. Besides, the RGD peptide could help achieve active tumor targeting via receptor-mediated endocytosis [57].

A sequential targeted mesoporous silica nanoparticle (MSN) encapsulating doxorubicin hydrochloride (DOX·HCl) was synthesized by conjugating tumor-shreddable hyaluronic acid (HA) on the surface of MSNs via disulfide bonds (MSNs/SS/HA@DOX). The MSN was used in CD44-targeted and redox-responsive drug delivery system. DOX·HCl was the model drug in the system. HA, a CD44 ligand, enabled the MSNs to have a higher cellular uptake efficiency by HeLa cells, which over-express CD44, and then leading CD44-mediated endocytosis. The cellular uptake of MSNs/SS/HA@DOX to HeLa cells was greater than that of LO2 cells, which were CD44 deficient [59]. In another study, the mesoporous silica nanoparticles (MSNs) were modified with folic acid (FA) and dexamethasone (DEX). Thiol groups and amine groups were modified on the surface of the MSNs. Thiol groups were used to conjugate DEXs, while amine groups were used to conjugate the FAs. The model drug, DOX, was set in the inner part of the MSNs. Folic acid can bind the folate receptor on the cancer cell surface for active tumor cell targeting and allow receptor-mediated endocytosis. DEX is a potent glucocorticoid. It could bind to the nuclear receptor, glucocorticoid receptor (GR), expressed in almost every cell type to achieve active nuclear targeting [23].

A supramolecular nanomedicine with an efficient nuclear accumulation of dual anticancer drugs was reported. The nanomedicine was loaded with cisplatin and 10-hydroxycamptothecine (HCPT). Cisplatin could inhibit DNA synthesis by interacting with DNA. HCPT is the DNA topoisomerase I inhibitor. The synergistic effects were observed when topoisomerase I inhibitors were combined with cisplatin in several tumor cells [24]. However, the HCPT has difficulties entering the nucleus. This issue could be solved by co-delivery with cisplatin, a drug with a positive charge. Cisplatin assembles HCPT and helps the dual drugs enter the nucleus [24].

### 2.3. Aptamers Nuclear Targeting

Aptamers are short single-stranded RNA or DNA molecules that can be artificially made. Aptamers usually consist of 20 to 60 nucleotides and have high specificity and affinity, which enable them to bind with the targeting cells [26]. Their high affinity makes them similar to antibodies. However, the aptamers have lower immunogenicity and toxicity, making them less likely to develop drug resistance [25,55]. Aptamers can be used as the drug delivery carriers for subcellular level targeting. To achieve sequential targeting, an anticancer drug, was conjugated with aptamers to form an aptamer-drug conjugate (ApDC). APTA-12, AS1411 and APTA-12 aptamers have dual modes of action by acting as anticancer agents and binding to nucleolin. Nucleolin is distributed in the cell membrane, cytoplasm, nucleus and nucleolus [60]. Cell surface nucleolin could mediate ligand internalization in a calcium-dependent manner. Inside of the cell, nucleolin could serve as a shuttling protein between the cytoplasm and the nucleus [61].

The APTA-12 drug complex could overcome AS1411′s poor cellular penetration, immunogenicity, and non-specific toxicity. It showed good chemotherapeutic effects against pancreatic cancer [25]. In one study, DOX-AS1411 aptamer complex (Ap-DOX) was formed through noncovalent intercalation. The complex could be directly delivered to the cancer cell nucleus by using the shuttling properties of nucleolin. The Ap-DOX complex was loaded in the liposome to protect the complex from disintegration. After the liposomes were taken up by the cancer cell, the Ap-DOX complex would actively transport to the nucleus since the aptamer could interact with nucleolin. The DOX is well protected during the whole process [55]. In addition, Joshi et al. showed that the APTA12-DOX conjugate could also increase cancer cell selectivity and had better cytotoxicity than the naked drug. The APTA12-DOX conjugate could reduce DOX accumulation in non-target cells and decrease non-specific binding and potential cardiotoxic side effects [26].

### 2.4. Zwitterionic Carbon Dots Nuclear Targeting

The pH values are different between normal tissue and tumor microenvironment. Normal tissue’s pH is about 7.4, while tumor cell tissue’s pH is around 6.8 [54]. Meanwhile, the intracellular pH values are different between cytosol and the nucleus. The nucleus’s pH is consistently 0.3–0.5 units higher than cytoplasm’s [27]. These characteristics provide an idea of using the pH differences to achieve specific cancer cell targeting and nucleus-controlled drug release.

Carbon dots (CDs) have biocompatibility and possess bright fluorescence. CDs can load aromatic drugs through π–π interactions [62]. The fluorescence imaging studies showed that CD could enter cancer cells faster than normal cells, and after internalization, the CD could enhance anti-cancer drug nuclear delivery [27,63,64]. It is a suitable drug carrier in the drug delivery system. After being modified with specific molecules or components, it became a zwitterionic carbon dot, used in a pH-sensitive drug delivery system [27].

A β-alanine was introduced to the CDs to serve as the zwitterionic functional group. The particle could alter its surface charge according to the environmental pH. The nanoparticles in the tumor microenvironment are positively charged and are internalized by the cells first. When the particles approach the nucleus, they are negatively charged and interact with the nuclei. Constructed by non-covalent grafting of DOX, the CD-based delivery system showed better nuclear accumulation. The CD performed superior antitumor efficacy and resulted in tumor growth inhibition. The zwitterionic CDs also have high biocompatibility and can effectively translocate into the nucleus. They are resistant to non-specific protein adsorption and have good colloidal stability over a wide pH range [27].

In another study, the zwitterionic CDs were wrapped in the cell-derived microvesicles (MVs) as the pH-responsive traceable nanocarrier to achieve specific cancer targeting. The MVs’ natural composition and structure made them avoid being captured by the immune system and actively target the cancer cells. When the complex was in blood (pH = 7.4), the zwitterionic CDs were tightly bound with the DOX by electrostatic interactions and avoided premature drug leakage. Due to the fluorescence inner filter effect (IFE) between the drug and the CDs, CDs are not fluorescent in the blood. After entering the tumor cell, the low pH in the lysosome (pH = 4.5–5.5) triggered the CDs to reverse the charge and repulsed the DOX. Therefore, the fluorescence of DOX was recovered, and the drug release process was trackable [28].

## 3. Mitochondria Targeting

Mitochondria, a double membrane-bound organelle, is one of the most important organelles for energy generation and metabolic regulation in eukaryotes [65,66]. In mitochondria, oxidative phosphorylation (OXPHOS) produces ATP (adenosine triphosphate) that serves as an energy resource for maintaining cells’ operation, and most ROS (reactive oxygen species) are generated during the process of ATP synthesis [67]. Previous studies showed that mitochondria in cancer cells would alter the OXPHOS process into glycolysis which later causes reduction of ATP and some other changes [68]. The over-accumulation of ROS caused by the hypoxic conditions induces more electrons leaking from the electron transport chain. The membrane potential is altered due to the non-utilization of a proton gradient. The pH change happens with the production of lactic acid by glycolysis. These changes cause mitochondria dysfunction and later induce diverse diseases [69,70]. The Bcl-2 family is mainly located at the mitochondria outer membrane and plays an important role in controlling apoptosis at mitochondria. Bax and Bak, Bcl-2 proteins, are the core regulators of apoptosis that can penetrate the mitochondria outer membrane and induce the release of apoptotic factors like cytochrome c or SMAC/DIABLO from the mitochondria matrix to the cytosol. This process is considered irreversible leading to final death [71,72,73].

The importance of diverse mitochondria functions on cells promotes a cancer treatment strategy on establishing a mitochondria-targeting system. Drugs need to assemble with some active components, effectively delivering drug molecules to mitochondria. Then, the drug could accumulate in mitochondria and play their therapeutic effects to destruct cancer cells. These active moieties can be categorized according to their properties such as chemical structures, size, affinity, and charges. Here we summarize the sequential targeting system according to the mitochondria targeting moieties (Figure 2).

### 3.1. Lipophilic Cation

The constituent components and the membrane potential changes of mitochondria membranes are key factors to select suitable targeting moieties for a mitochondria targeting system [70]. First, the outer and inner mitochondrial membranes and the matrix between them make up the structure of the mitochondria membrane. The outer and inner membranes are composed of the phospholipid bilayer [69]. Second, as the proton pumps are used in ATP synthesis, a proton gradient is generated across the mitochondria membrane, and the inner membrane presents as negative membrane potentials [74]. Compared with the plasma membrane potentials, the potentials in the inter-membrane space (120–180 mV) are 3–5 times higher than the plasma membrane potentials, normally ranging from 30 to 40 mV. This discrepancy triggers the accumulation of cations around the mitochondria membrane and can be explained via the Nernst equation [75]. The membrane potentials of mitochondria (ΔΨ) at the temperature of 37 °C can be calculated as Theorem 1 follows:(1)$$△\Psi (\mathrm{mV})=61.5\times {㏒}_{10}\frac{c_{\mathrm{in}}}{c_{\mathrm{out}}}$$
where c*_out_* and c*_in_* are the concentrations outside and inside compartments of the inner mitochondria membrane. The units are all mV.

Therefore, lipophilic cations can be used as mitochondria targeting moiety in mitochondria targeting drug delivery systems. For example, triphenylphosphonium (TPP) cations, dequalinium (DQA), rhodamine, cyanine cations, quaternary ammonium salts, pyridinium, and berberine cations [36,70].

TPP cations and DQA derivatives are the most common choices in mitochondria targeting systems. They can target the mitochondria by their strong lipophilicity and membrane potential properties [35]. DOX and PTX are often used in subcellular targeting therapy research because of their multiple anticancer effects. They can be conjugated with lipophilic cations through chemical reactions [76]. Some studies indicated that lipophilic cations could be used in sequential cancer mitochondria targeting therapy. For example, TPP cations were grafted on the surface of silica particles, and DOX was encapsulated into the pores. The complex was coated with HA, a ligand for the CD44 receptor. The mesoporous silica nanoparticles could enter cancer cells via CD44 receptor-mediated endocytosis with the help of HA and later accumulate in mitochondria with the help of TPP cations [35]. Another study used TPP and HA to coat polypyrrole-silica nanoparticles that were modified with (3-aminopropyl) trimethoxysilane (APTMS). DOX was encapsulated in the nanoparticle. After the nanoparticles entered the cells, they were gathered in mitochondria and killed the cancer cells under the assistance of NIR laser irradiation, which worked as the hyperthermia component [36].

DQA is another lipophilic cation that can enhance drug accumulation in mitochondria. Tian et al. utilized DQA’s lipophilic property and delocalized cationic nature to coat PTX-loaded liposomes via thin-film hydration. HA was also used to coat the compound by electrostatic adsorption. The nanoparticles can enter cancer cells via HA-CD44 interaction and then aggregate in mitochondria under the assistance of DQA [37]. All these studies showed a higher inhibitory effect on cancer cells and the possibility of overcoming multi-drug resistance. Therefore, the lipophilic cations can be used for conjugation with other biologically active components in cancer sequential mitochondria targeting therapy systems.

### 3.2. Dual Targeting to Mitochondria and Cell Membrane Moiety

Glycyrrhetinic acid (GA) is the main constituent of *Glycyrrhiza glabra* [77]. It can express its antioxidant or pre-oxidation effect depending on the target type. The pro-oxidation effect can be acted through interaction with the transition metals, especially the Fe^+^ in the mitochondrial respiratory chain. GA was reported to process anti-cancer activity and induce apoptosis via PPAR-γ activation and NF-κB inactivation [78]. GA can also induce the mitochondria-mediated apoptosis (MMA) pathway by opening the mitochondrial permeability transition pore and decreasing mitochondria’s membrane potential [79]. GA was found to recognize cell surface membrane protein kinase C (PKC) α to exert expression functions via its active site, like the C_30_-carboxyl group [38,80,81]. These properties indicate that GA is a good candidate in sequential targeting mitochondria therapies.

Zhang et al. designed one GA-based sequential drug delivery system. They conjugated GA with functionalized graphene oxide (GO) nanoparticles. DOX was used as the model drug. After entering the cell, the lipophilic nanoparticle was targeted to mitochondria via the help of GA. This study presented lower toxicity of the drug delivery system than the non-GA conjugated system [38]. In another study, GA was conjugated with DOX onto N-(2-hydroxypropyl) methacrylamide (HPMA) copolymer backbone by hydrazone bond. Then, they were attached to the surface of gelatin, a substrate of metal matrix protease-2 (MMP-2), to form the GNPs-P-Dox-GA nanoparticles. The GNPs-P-Dox-GA nanoparticles were degraded by extracellular MMP-2. The nanoparticles were digested in smaller sizes that enhanced tumor tissue penetration. Then, the nanoparticles entered the cell by the mediation of GA and efficiently targeted to mitochondria after detaching from HPMA copolymer via hydrazone bond hydrolysis. This study showed a higher accumulation of GA conjugated nanoparticles in mitochondria than the naked one. Meanwhile, the drug delivery system could overcome drug efflux, which affects the efficiency of drug therapies [39]. Therefore, GA shows an excellent application in both cellular and subcellular targeting. It can be well utilized in sequential targeting systems to overcome multidrug resistance and enhance drug accumulations in specific sites with lower toxicity.

### 3.3. Peptide

Mitochondria-penetrating peptides (MPPs) could be effectively uptaken by diverse cells. The peptide sequences were synthesized by considering specific chemical properties for achieving specific binding with different organelles. MPPs are designed to be both cationic and lipophilic, which can target mitochondria via reasonable controls of their lipophilicity and charge [82]. As a result, MPPs can be considered one promising category in cancer sequential targeting systems.

Recently, Szeto–Schiller (SS) peptides represent a novel approach for mitochondria targeting drug delivery systems. A series of SS-peptides has been synthesized, and four of them (SS-01, SS-02, SS-20, and SS-31) were extensively studied. Among them, SS-31 presented excellent potential in mitochondria targeting compared with other SS peptides. SS-31 contains Dmt (2′6′-dimethyltyrosine) that showed greater scavenging ability since it retains the aromatic–cationic motif rather than putting Dmt as the N-terminal amino acid [83,84]. Cardiolipin is a special phospholipid expressed on the inner mitochondria membrane (IMM). SS-31 can accumulate and selectively bind to cardiolipin by electrostatic and hydrophobic interactions with the IMM. It can help improve mitochondrial bioenergy by altering membrane properties and activity modification in the electron transport chain (ETC) protein complexes via cardiolipin interactions [84,85].

Liu et al. formulated SS-31 loaded HA-chitosan nanoparticles via electrostatic interaction. The nanoparticle could be selectively taken up by CD44 overexpressed cells. Then, the nanoparticles were broken up in the low pH environment of the lysosomes and released SS-31, which subsequently targeted mitochondria for therapeutic effects. The complexity showed better anti-oxidative and anti-apoptotic effects than the free SS-31 [40]. Another research demonstrated a cancer sequential targeting therapy system using photostable nanodiamonds as carriers. The carriers conjugated with mitochondrial localizing sequence (MLS) peptides and FA could target the folate receptors on the surface of most cancer cells. The nanosystem-loaded DOX exerted a sequential targeting effect and a better cellular uptake of DOX in doxorubicin-resistant experiment models [41]. Therefore, mitochondria-targeted peptides also can be a meaningful research direction to overcome the difficulty of diffusive transport in the inner mitochondria membrane.

## 4. Endoplasmic Reticulum/Golgi Apparatus Targeting System

The endoplasmic reticulum (ER) and Golgi apparatus are critical membrane-bound organelles participating in protein synthesis, processing, and transportation [86]. The dysfunction of the ER/Golgi network can initiate cell death which attracts increasing interest in developing the ER-targeted drug delivery systems for cancer treatment [87]. Nanoparticles with specific properties can vector the loaded drug to tumor cells via endocytosis. Kang et al. constructed a cell-penetrating peptide (CPP)-conjugated lipid/polymer hybrid nanovesicles (LPNVs) for ER targeted delivery. Penetratin, the CPP, could enhance target cell internalization via combined electrostatic interaction-driven cell adhesion and ATP-dependent endocytosis. Then, penetratin delivered the nanoparticle specifically to ER prior to biodegradation [47]. Luo et al. developed a chondroitin-modified targeted drug carrier system to treat liver cancer and liver fibrosis via CD44-mediated endocytosis [49,50]. The nanoparticle loaded with DOX and retinoic acid could accumulate in the Golgi apparatus via interaction with N-acetylgalactosaminyltransferase (GalNAc-T) and trigger Golgi collapse to induce apoptosis.

So far, most studies of ER–Golgi-targeted nanomedicines mainly focus on the subcellular level [88,89], while the sequentially active-targeted drug delivery systems for cancer treatment remain challenging.

## 5. Conclusions and Perspectives

To date, many studies have investigated the sequential targeted drug delivery system for cancer treatment. Due to the dual-targeting effect, sequential active-targeted drug delivery systems can improve the anticancer therapeutic efficiency and reduce the adverse effects. This review spotlighted different targeting strategies categorized by the final targeting organelles.

Four nuclear targeting mechanisms were discussed, including proteins/peptides, small molecules, zwitterionic carbon dots and aptamers. Proteins and peptides are biocompatible with cells. They can serve as drug delivery carriers to achieve specific targeting at the cellular level [51]. If they were modified on the carrier surface as a targeting moiety, the peptide could deliver the drug molecule to the nucleus [21,22]. Small molecules can also be used as the targeting moiety and modified on the carrier surface to help cellular and subcellular targeting [59]. In addition, zwitterionic nanoparticles can control drug release depending on the different pH values between cytoplasm and nucleus [28]. Besides, aptamers are artificially synthesized nucleotides and have a high affinity to their targets. This property makes them a suitable carrier in sequential drug nuclear targeting, and the development of novel aptamers for specific targeting is in urgent need [25]. Nevertheless, if the nuclear targeted particle loses specificity and accumulates in normal cells, higher side effects would be expected.

Lipophilic cations, glycyrrhetinic acid, and peptides have been used for anticancer treatments for the sequential mitochondria targeting system. They could help achieve better treatment effects via the advantages of specific cancer cell mitochondria accumulation and overcoming drug resistance [69,70,90]. However, more research is still needed to confirm the thorough mitochondria targeting therapy mechanisms. Besides, the lack of specificity of malfunction mitochondria targeting also needs to be solved. Meanwhile, the development of mitochondria targeting components used for neurodegenerative disorders is in urgently needed. Those targeting components need to exert functions via improving the metabolism and preventing apoptosis of neural cells, which is different from cancer cells [91]. Hence, the development of those mitochondria targeting moieties will be an important research direction in the future.

Compared with the nucleus/mitochondrial-targeted therapeutics, studies on the sequential ER/GA-targeted drug delivery systems are still rare, mainly because the targeting mechanisms remain unclear [87]. We believe this review will offer valuable information for researchers to design a better sequential cell organelle-targeting drug delivery platforms. However, there are still many issues that need to be solved. First, a detailed understanding of sequential transport and therapeutic mechanism of the targeted nanomedicines is required. Second, its clinical application is still hampered, and further in-depth studies are needed to reach clinical trials. Although the study of sequential targeted drug delivery systems is still at an early stage, it holds a great promise of precision medicine for cancer.

## Figures and Tables

**Figure 1 pharmaceutics-14-00573-f001:**
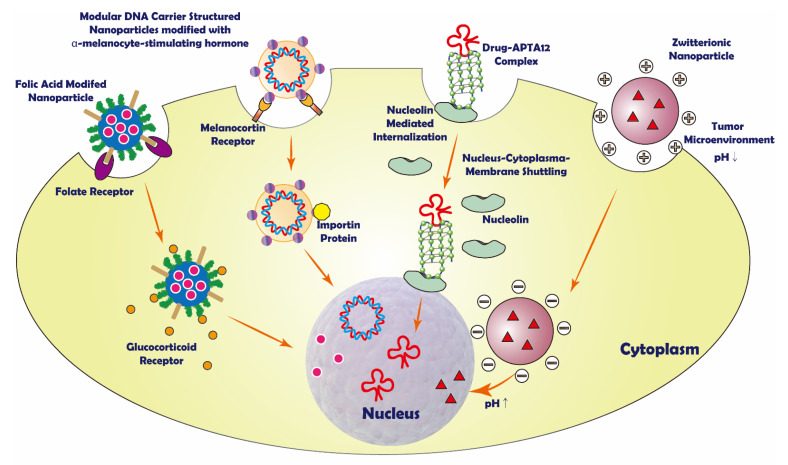
Schematic representation of some sequential nuclear targeting strategies.

**Figure 2 pharmaceutics-14-00573-f002:**
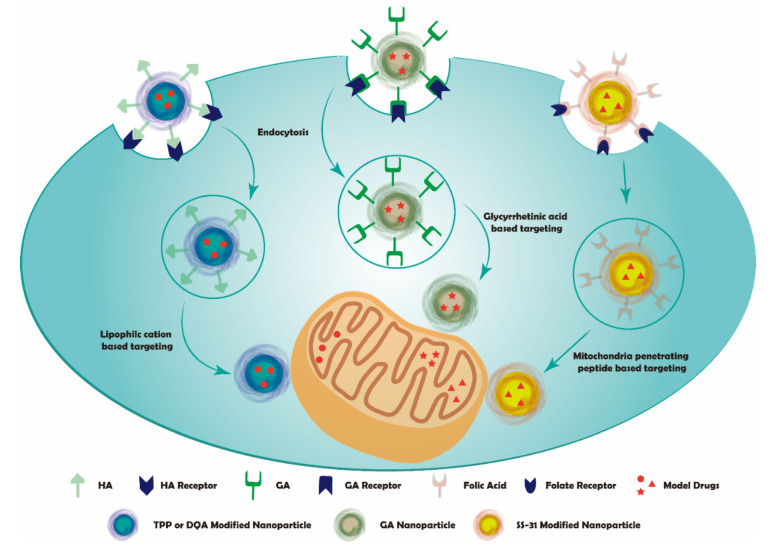
Schematic representation of sequential mitochondria targeting.

**Table 1 pharmaceutics-14-00573-t001:** Some physical and bioactivity characteristics of sequential-targeting drug delivery systems.

NP Matrix	Drug	Targeting Moieties	Cellular Targeting	Subtargeting Moieties	Target Organelles	References
liposomes	DOX	transferrin	transferrin receptors	octaarginine	nucleus	[21]
silica	DOX	galactose	galactose ligands	TAT peptides		[22]
silica	DOX DEX	folic acid	folate receptors	DEX-GR complex		[23]
peptide	HCPT and cisplatin	RGD peptide moiety	integrin	positive charge		[24]
aptamer	gemcitabine	AS1411	nucleolin	AS1411		[25]
aptamer	DOX	APTA12	nucleolin	APTA12		[26]
carbon	DOX	positive charge	phosphate	negative charge		[27,28]
polymer	apoptin	positive charge	negative charge	HKRRR		[29]
polymer	GNA002	cRGD	Integrin ανβ3-receptor	hexa-arginine		[30]
peptide	siRNA and NLS with influenza virus hemagglutinin	AS1411	nucleolin	NLS peptide		[31,32]
peptide	camptothecin	HNLS-3	negative charge	HNLS-3		[33]
polymer	9-nitro-20(S)-camptothecin	HA	CD44 receptor	TAT peptides		[34]
silica	DOX	HA	CD44 receptor	TPP	mitochondria	[35]
silica	DOX	HA	CD44 receptor	TPP		[36]
lipid	paclitaxel	HA	CD44 receptor	DQA		[37]
graphene oxide	DOX	GA	GA receptor	GA		[38]
gelatin	DOX	GA	GA receptor	GA		[39]
polysaccharide	SS-31 peptides	HA	CD44 receptor	SS-31 peptides		[40]
nanodiamonds	DOX	folic acid	folate receptor	MLS peptides		[41]
polymer	siBcl-xL	HA	CD44 receptor	KLA		[42]
silica	5-fluorouracil	HA	CD44 receptor	coumarin		[43]
lipid	cyclopeptide RA-XII	HA	CD44 receptor	TPP		[44]
metal	DOX	folic acid	folate receptor	TPP		[45]
polymer	DOX	folic acid	folate receptor	TPP		[46]
lipid/polymer hybrid nanovehicles (lpnvs)	N/A	penetratin	electrostatic interaction between the LPNV and cells	penetratin	endoplasmic reticulum	[47]
polymer	Ru-1	biotin	biotin receptor	tetraphenylporphyrin		[48]
lipid	DOX + retinoic acid	chondroitin sulfate	CD44 receptor	chondroitin sulfate	Golgi apparatus	[49,50]

DOX: doxorubicin; PTX: Paclitaxel; DEX: dexamethasone; HCPT: 10-hydroxycamptothecine; RGD: Arg-Gly-Asp; HA: hyaluronic acid; GA: glycyrrhetinic acid; TPP: triphenylphosphine; DQA: dequalinium; DEX-GR complex: dexamethasone-glucocorticoid receptor; APTA12: gemcitabine incorporated G-quadruplex aptamer; NLS: nuclear localization signal; HNLS-3: PFVYLIPKKKRKVHHHHHHGC-NH_2_; MLS: mitochondrial localizing sequence; siBcl-xL: small interfering RNA (siRNA) against Bcl-xL.

## Data Availability

Not applicable.

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
