# Peer review of "Sequential Drug Delivery in Targeted Cancer Therapy"

_pharmaceutics, 2022, doi:10.3390/pharmaceutics14030573_

Round 1
Reviewer 1 Report
The review is clear, well organized and offers the latest information in the field. I do not have comments to add to this manuscript.
Author Response
The authors would like to thank the reviewer’s comment.
Reviewer 2 Report
This is an excellent review that should be published in Pharmaceutics. I have very minor commentsQ
- On the abstract, it is better to say: Cancer is a major public health problem and one of the leading cause of death...
- On the abstract, it would be better to mention the subcellular targets (nucleus, mitochondria, etc.
- It will be good to mention the endosmolytic materials that rupture the endosome through the "proton sponge mechanism" when the nanoparticles enter the cytoplasm on page 4 on endocytosis.
- Concerning glytcyrrhetinic acid it would be good to mention a review that mentions how it targets cancer cells such as this this one doi: 10.2174/0929867323666160112122256.
I think that after these minor corrections it can be published to Pharmaceutics.
Author Response
This is an excellent review that should be published in Pharmaceutics. I have very minor commentsQ
- On the abstract, it is better to say: Cancer is a major public health problem and one of the leading cause of death...
Response: The statement has been changed (line 11).
- On the abstract, it would be better to mention the subcellular targets (nucleus, mitochondria, etc.
Response: The statement has been added to the abstract (lines 16-18).
- It will be good to mention the endosmolytic materials that rupture the endosome through the "proton sponge mechanism" when the nanoparticles enter the cytoplasm on page 4 on endocytosis.
Response: The “proton sponge mechanism” has been added to lines 86-87.
- Concerning glytcyrrhetinic acid it would be good to mention a review that mentions how it targets cancer cells such as this this one doi: 10.2174/0929867323666160112122256. 26758798
Response: This review has been added to lines 307-308.
I think that after these minor corrections it can be published to Pharmaceutics.
Reviewer 3 Report
Dear Editor,
I have carefully read the manuscript entitled "Sequential Drug Delivery in Targeted Cancer Therapy" by Han Yu et al. that deals with intracellular delivery of actives against tumors.
Even if there are many examples reported, the authors should address other issues in cancer targeted theraphy that are ignored.
For instance, in the introduction, the authors should consider to insert a discussion on the main problem of anticancer targeted therapies, that is that EPR effect in humans doesn't exist!
The main question here is how to arrive inside the parenchyma before discussing how to reach intracellular cancer cell compartpents.
Secondly, I suggest to add some references on nuclear drug delivery by 0-D carbon nanodots.
Some example is reported in fellows:
1) ,
Author Response
Dear Editor,
I have carefully read the manuscript entitled "Sequential Drug Delivery in Targeted Cancer Therapy" by Han Yu et al. that deals with intracellular delivery of actives against tumors.
Even if there are many examples reported, the authors should address other issues in cancer targeted theraphy that are ignored.
For instance, in the introduction, the authors should consider to insert a discussion on the main problem of anticancer targeted therapies, that is that EPR effect in humans doesn't exist!
The main question here is how to arrive inside the parenchyma before discussing how to reach intracellular cancer cell compartpents.
Response: The discussion has been added to lines 44-47.
Secondly, I suggest to add some references on nuclear drug delivery by 0-D carbon nanodots.
Some example is reported in fellows:
1) ACS Applied Materials and Interfaces, 2022, 14(2), pp. 2551–2563;
2) Cancers, 2020, 12(11), pp. 1–23, 3114;
3) Scientific Reports volume 5, Article number: 18807 (2015)
Response: These references have been added to lines 207-209.